# Metagenomic Sequencing Reveals that High-Grain Feeding Alters the Composition and Metabolism of Cecal Microbiota and Induces Cecal Mucosal Injury in Sheep

Fei Xie,[a,b] Lei Xu,[a,b] Yue Wang,[a,b] Shengyong Mao[a,b]

aLaboratory of Gastrointestinal Microbiology, College of Animal Science and Technology, Nanjing Agricultural University, Nanjing, China
bNational Center for International Research on Animal Gut Nutrition, National Experimental Teaching Demonstration Center of Animal Science, Nanjing Agricultural University, Nanjing, China

**ABSTRACT** The cecum serves as an additional fermentation site for ruminants, but it lacks buffering capacity and has a relatively simple epithelial structure compared to the rumen. The role of high-grain (HG) diets in manipulating the rumen microbiome has been well elucidated, yet the microbial response to such diets in the cecum and the subsequent microbe-host interactions remain largely unexplored. Here, we describe the modification of the cecal microbiome and host epithelial gene expression based on data from 20 sheep grouped to feed an HG diet for 7, 14, and 28 days. Our data indicate that the alteration of cecal microbial fermentation was manifested by a decrease in luminal pH and an increase in acetate and butyrate concentrations following the diet change to HG. We further demonstrate that the alteration of the microbiome was driven by microbes that are likely acetate producers (e.g., *Blautia* spp. and *Akkermansia* spp.) and butyrate producers (e.g., *Anaerostipes* spp. and *Roseburia* spp.). Moreover, the core microbiota in the cecal microbiome was predominantly maintained after HG diet feeding, while the specific populations of the cecal microbiomes adaptively varied at the species and genomic levels time dependently. Association analysis suggests that the perturbations of the cecal microbiome under the HG diet were closely linked to the variations in the two key enzymes that catalyze the conversion of pyruvate to acetyl-CoA and urease enzymes that hydrolyze urea into ammonia, alongside mucosal inflammatory responses. Overall, our findings here provide novel insights into understanding microbiome-host interactions in the hindgut of ruminants.

**IMPORTANCE** High-grain (HG) diets are known to alter the rumen microbiome. However, the responses of the hindgut microbiota and its epithelial function need further investigation in ruminants. Using 20 sheep as the experimental model, we found that the microbial fermentation pattern of the cecum changed after switching to the HG diet. The taxa of the acetate and butyrate producers increased with the feeding time. Moreover, enzymes engaged in carbon and nitrogen metabolisms of the cecal microbiome are altered. The expression of epithelial genes related to volatile fatty acid (VFA) absorption and metabolism, cytokines, and tight junction proteins, alongside light microscopy visualization of epithelial tissue, suggested that the HG diet may induce cecal mucosal inflammatory responses. Our findings reveal cecal microbial and metabolic perturbations in response to HG diets in sheep and provide a new reference for the research on hindgut microbial homeostasis and host health in ruminants.

**KEYWORDS** sheep, cecal microbiota, high-grain diet, microbial perturbation, epithelial function, microbiome-host interactions

Address correspondence to Shengyong Mao, maoshengyong@njau.edu.cn.

Ruminant animals possess an ecologically diverse gut ecosystem comprising a complex microbial community that is well recognized as a crucial contributor to generating metabolites for energy production from low-quality forages, ultimately for milk and meat production (1). The rumen is a specialized fermentation chamber in the foregut of ruminants (2). The microbial processes involve the fermentation of sugar, starch, cellulose, hemicellulose, and pectin into $CO_2$, methane, $H_2$, and volatile fatty acids (VFAs) (3–5). However, there is increasing evidence that the hindgut and its resident microbiota are imperative for ruminant health and efficient production. Compared to monogastric animals, this relationship has not been extensively studied to date (6, 7). Therefore, understanding the interactions between hosts and microbes in the hindgut and their implications for animal health and production performance is important for ruminants.

The cecum is one of the main fermentation sites in the hindgut of ruminants. It can ferment approximately 17% of the digested cellulose and provide 12% of the total VFAs for the animal (8). Compared with the rumen, fermentable substrates in the cecum are limited to slower-digesting polymers such as lignin and crystalline starches that escape foregut digestion and absorption, alongside some secreted mucins (2). However, the cecum lacks the buffering capacity of protozoa and saliva and has a different epithelial structure, which may make it less capable of maintaining the lumen pH (9).

High-grain (HG) diets are often fed to animals to increase milk production and growth performance, but excessively fermentable carbohydrates can result in a decrease in ruminal pH, an increase in epithelial permeability, and changes in the composition of VFA and fermentation processes (10, 11). The aggregation of these events could eventually induce subacute ruminal acidosis (SARA), a severe metabolic disorder in ruminants (12). Previous studies have reported dramatic changes in the rumen environment, including the microbial population and metabolites, during SARA, producing a negative impact on the animal's performance (13, 14). When dietary factors contribute to relatively larger proportions of grains and smaller proportions of forage (i.e., excessive flow of fermentable carbohydrates from the small intestine), hindgut acidosis can occur (9). Events that appeared in the rumen during SARA are mirrored in the hindgut. We previously studied the dynamic shifts in the composition of mucosa-associated microbiota and epithelial health in the colon of sheep fed an HG diet (15). However, the alteration in the composition and function of the microbiota in the cecal digesta and its interactions with the host during HG diet feeding remain poorly understood.

Here, we investigated how a shift in the diet alters the hindgut microbiome by sampling the sheep cecum after different lengths of time of switching to an HG diet. We sequenced the 16S rRNA gene and used shotgun metagenomic sequencing to provide in-depth insight into the hindgut microbial adaptation response to the challenge of a diet shift. In most situations, hindgut acidosis is accompanied by an inflammatory response (16). Thus, potential effects on the gene expression of the epithelial tissue were also determined using real-time quantitative PCR, analyzing transcripts involved in VFA absorption alongside metabolism, cytokines, and tight junction proteins.

## RESULTS

**Shifts in fermentation parameters of cecal microbiota in sheep.** Our previous research found that feeding an HG diet enhanced the average daily gain of sheep and increased the VFA concentration in the rumen (11). Here, we aimed to determine the supplementary fermentation that occurs in the sheep cecum, which supplies the host with energy and nutrients by fermenting complex lignocellulose and starch that have escaped foregut digestion. By comparing animals that were given the hay (CON [control] group) and HG (HG7 [HG diet for 7 days], HG14, and HG28 groups) diets, we found that the pHs of cecal digesta decreased in the HG-fed animals (Fig. 1A; see also Table S1 in the supplemental material). The concentrations of the total VFAs, acetate,

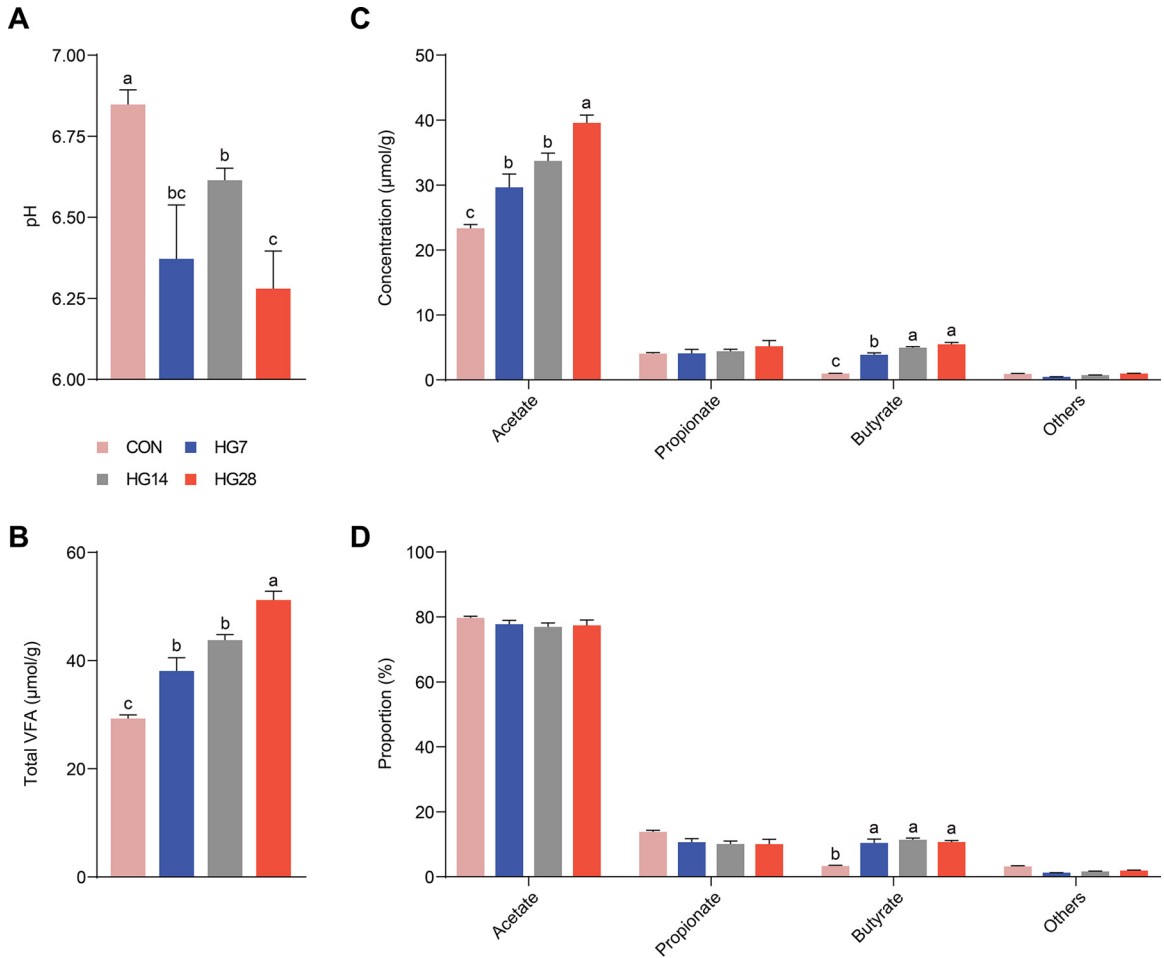

**FIG 1** Fermentation parameters detected in the cecal digesta of sheep. (A and B) Comparisons of the pHs (A) and the concentrations of total VFAs (B) in digesta before and after HG diet feeding with various feeding times (on days 7, 14, and 28). (C and D) Comparisons of the concentrations (C) and proportions (D) of acetate, propionate, and butyrate (remaining grouped as "others") in digesta among the four groups. Different letters represent significant differences among the four groups ($P$ value of <0.05 by a Wilcoxon rank sum test).

and butyrate increased continuously after the change from the hay to the HG diet (Fig. 1B and C; Table S1). Acetate was the most common VFA in all the studied animals (77.9% of the total VFAs), followed by propionate (11.1%) (Fig. 1D; Table S1). The relative butyrate level increased from 3.3% for the hay-fed animals to 10.8% for the HG-fed animals, with an increase in the butyrate/acetate ratios from 0.04 to 0.14 (Table S1). Valerate represented a low proportion of the total VFAs and showed a dynamic change, decreasing on day 7 and increasing on day 28 (Table S1). These results indicate that the fermentation pattern of the hindgut changes when sheep are fed an HG diet, which likely relates to their microbiome perturbations.

**Effects of the HG diet on the microbial diversity of cecal digesta and mucosa.** To illustrate the effect of a diet shift on hindgut microbial diversity, we conducted a principal-coordinate analysis (PCoA) based on 16S rRNA gene analysis with an analysis of molecular variance (AMOVA) (Fig. 2A; Fig. S1A), which clearly revealed the dynamic divergence of the microbial community structure of the digesta and mucosa among the four groups. Rarefaction curves approached asymptotes across all animals and treatments, implying that the sequencing depth allows an effective standardization of biodiversity measures (Fig. S1B). We found that the $\alpha$-diversity of the microbiota in both the digesta and mucosa was reduced after feeding the HG diet, and the $\beta$-diversity compared to the CON group was the lowest on day 14 between the three time

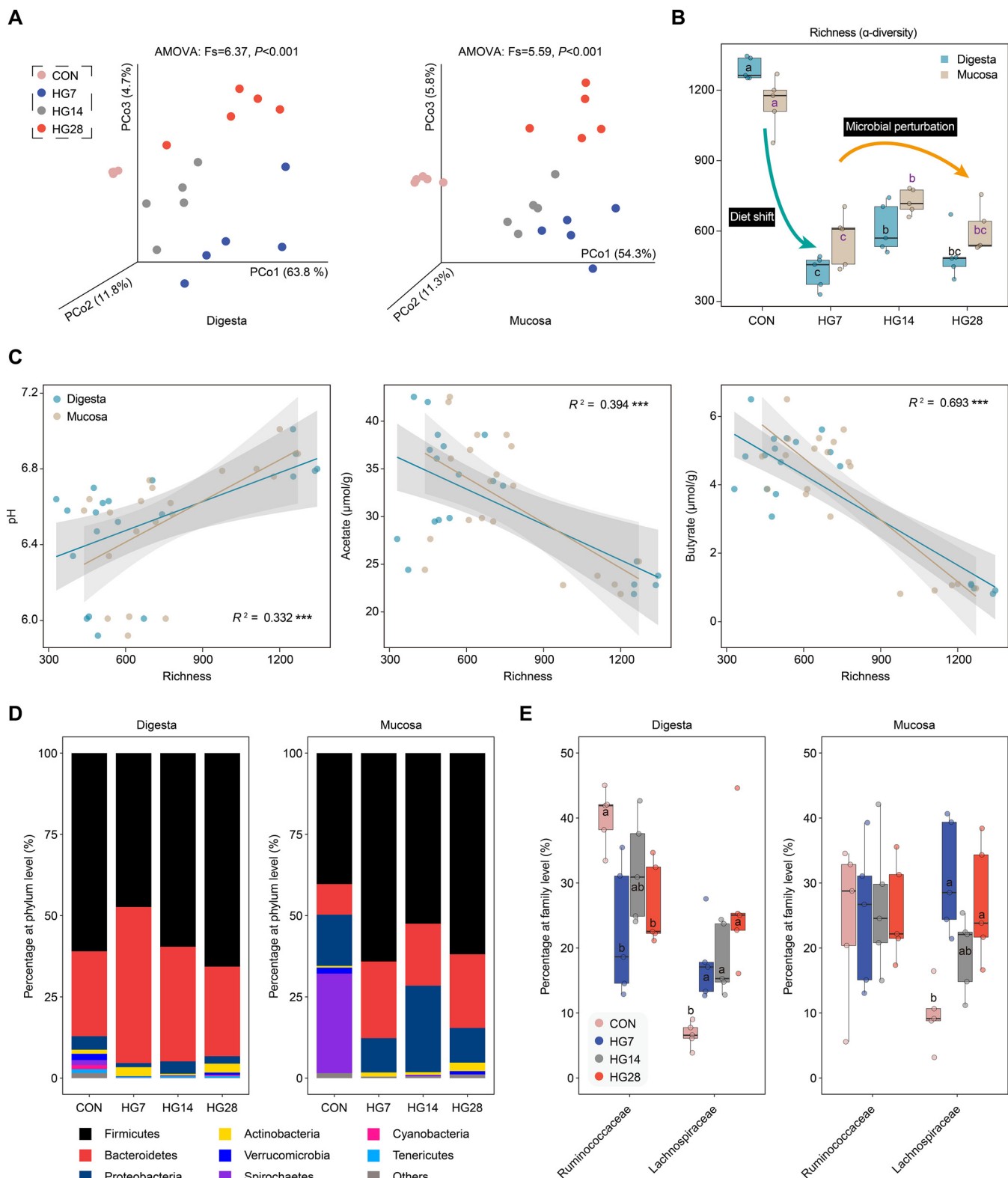

**FIG 2** Relationship between microbial structure and diet shift. (A and B) Changes in the microbial diversity of the bacterial community in the cecal digesta and mucosa before and after HG diet feeding, based on principal-coordinate analysis (A) and Chao 1 diversity analysis (B), respectively. (C) Regression lines showing the relationships between microbial composition indices and the pH and VFA concentration (acetate and butyrate). Significance is designated by using an *F* test for testing linear regression coefficients (***, $P < 0.001$). (D) Changes in the relative abundances of bacterial phyla in cecal digesta and mucosa samples among the four groups. (E) Comparisons of the variations of the dominant families (*Ruminococcaceae* and *Lachnospiraceae*) in the *Firmicutes* phylum after the introduction of the HG diet. Different letters represent significant differences among the four groups (adjusted *P* value of <0.05 by a Wilcoxon rank sum test).

points (Fig. 2B; Fig. S1C and Table S2A). Comparing the unweighted UniFrac distances between the CON and the three HG groups, we found that the variations in the digesta samples exceeded those of the mucosa (Fig. S1C), suggesting that the HG diet particularly contributed to a large number of shifts in the digesta. The lumen pH and VFA concentration were then linked to microbial community diversity (Fig. 2C), and we found that the microbial richness of the digesta and mucosa both positively correlated with pH and negatively correlated with acetate and butyrate concentrations. These results suggest that a diet shift drives significant changes in the cecum microbiota and that this impact was likely associated with variations in the pH and VFA concentrations.

When considering the microbial structure between the two diets (Fig. 2D; Table S2B), the phylum *Bacteroidetes* was more abundant in both digesta- and mucosa-associated communities during HG diet feeding. Upon the introduction of the HG diet, the relative abundances of the phylum *Firmicutes* decreased in the digesta and increased in the mucosa (Fig. 2D). However, there were no significant differences in the relative abundances of the phylum *Firmicutes*, which was inferred due to the opposite trend occurring between the families *Ruminococcaceae* and *Lachnospiraceae* in this phylum (Fig. 2E; Table S2B). A decline in the *Ruminococcaceae* family and a rise in the *Lachnospiraceae* family were observed after the diet shift (Fig. 2E), which are two common roles for degrading complex plant material in the gut environment (17). Notably, the low-abundance bacteria in the digesta communities (e.g., *Verrucomicrobia*, *Spirochaetes*, *Cyanobacteria*, and *Tenericutes*) were absent after HG diet feeding, and the *Spirochaetes* phylum (mainly *Treponema* spp.) dominating in the mucosa-associated communities drastically decreased from 30.6% for hay-fed animals to <0.5% for HG-fed animals (Table S2B). These results suggest that the implementation of an HG diet for sheep caused a pronounced shift in the microbial composition of the cecum.

**The HG diet reshaped the specific populations of cecal microbial communities.** We next explored how the diet stimulated specific populations of the sheep hindgut microbiome using metagenomic data. We found that the abundance of *Prevotella* spp. associated with polysaccharide utilization increased in the first 2 weeks (Fig. S2 and Table S2C). During the HG feeding process, the abundance of *Bacteroides* spp. gradually decreased, while many acetate producers (e.g., *Blautia* spp. and *Akkermansia* spp.) and butyrate producers (e.g., *Anaerostipes* spp. and *Roseburia* spp.) were prominent in the cecum of HG-fed animals (Fig. S2 and Table S2C). Monitoring the temporal microbiome changes in the target species, we observed comparable effects on different *Prevotella*-affiliated populations belonging to *Prevotella ruminicola* and *Prevotella* sp. strain KHP1 (Fig. 3A). *Ruminococcus bromii* and *Ruthenibacterium lactatiformans*, two members of the *Ruminococcaceae* family related to polysaccharide degradation, increased rapidly on day 7 and maintained significant enrichment at subsequent time points (Fig. 3A). The abundance of *Akkermansia glycaniphila* was missing after HG diet feeding, while *Akkermansia muciniphila*, a potential probiotic in the gut, highly accumulated on day 28 (Fig. 3A). Moreover, other probiotics derived from *Bifidobacterium*, such as *Bifidobacterium pseudolongum* and *B. merycicum*, also accumulated on day 28 (Fig. 3A). In general, the abundance of *Alistipes* spp. is highly relevant in dysbiosis and disease. We found that *Alistipes senegalensis* was significantly enriched on day 28 (Fig. 3A). Overall, these results suggest that the microbiomes adaptively vary time dependently in response to HG diet feeding.

The term "core microbiome" was recently employed to describe taxa with high occupancy, which persisted in most of the samples (18). We investigated whether any species-level phenotypes occurred in each sampled individual to compose a core gut microbiota. A total of 301 core species have been identified, which were defined as having an average relative abundance in at least one group of ≥0.1% and being present in more than 60% of the samples in one group (Fig. 3B; Table S2D). The phylogenetics of the core species comprised two predominant phyla (*Firmicutes* [59.1%] and *Bacteroidetes* [29.6%]) (Fig. 3B; Table S2D). Most of the species belonged to the genera *Bacteroides* (11%), *Prevotella* (6.6%), *Alistipes* (6%), *Clostridium* (6%), and *Ruminococcus*

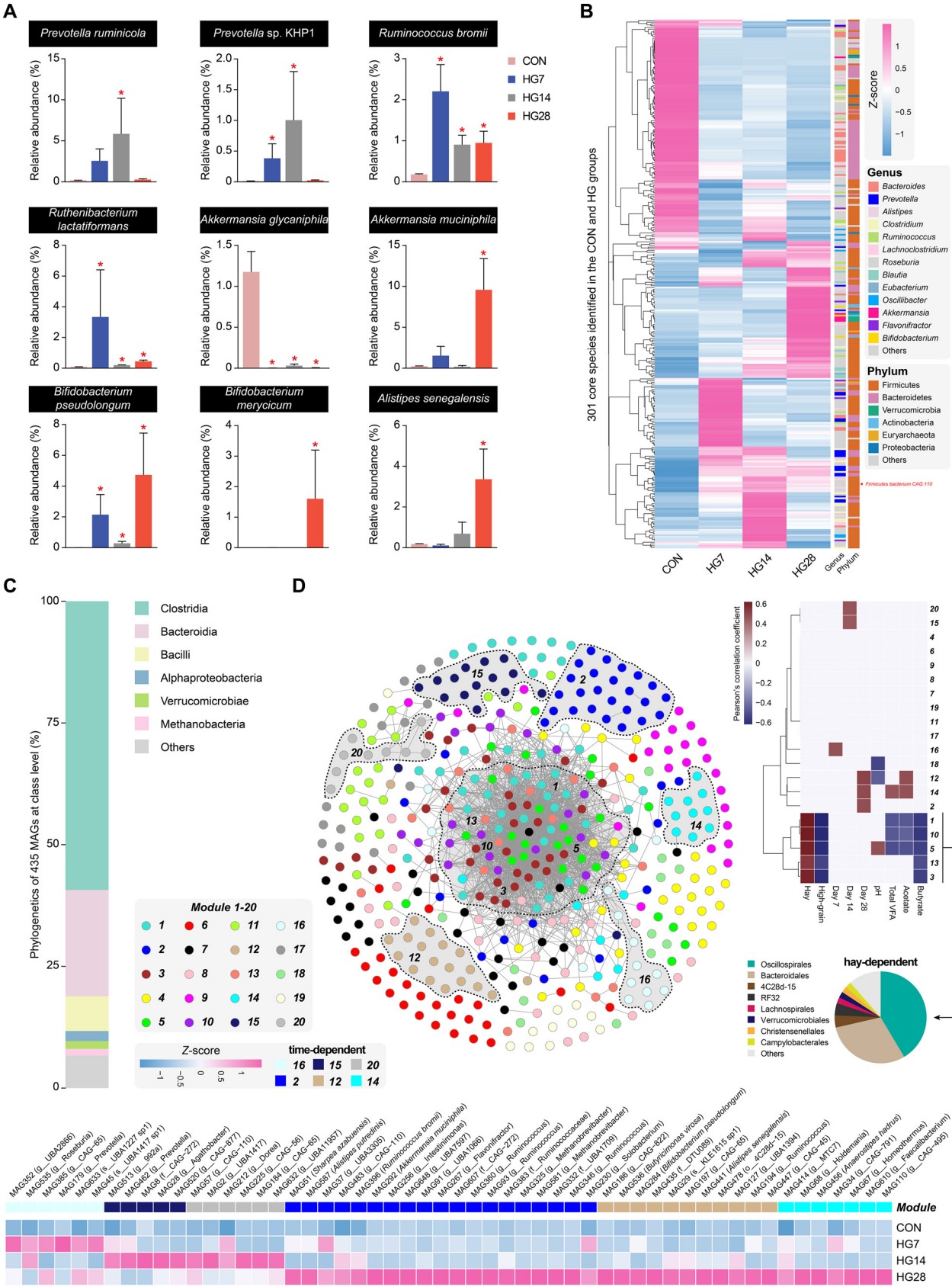

**FIG 3** Specific populations of cecal microbial communities. (A) The relative abundances of nine species were compared using the Wilcoxon rank sum test, with significant increases or decreases indicated between the CON and HG groups. *, adjusted *P* value of <0.05. (B) Heat map showing

(4%). We found that the abundances of these core species showed considerable variations in the four groups, while the total abundance remained stable, accounting for 60.2% of the microbial abundance in the CON group and 66.9%, 65.8%, and 72.2% in the three HG groups, respectively (Fig. 3B; Table S2D). Notably, *Firmicutes* bacterium CAG:110 displayed the highest abundance (average of 7.1% for the four groups) among these core species, and it was recently revealed to be a key species in the hindgut related to ethanolamine degradation and utilization (19). These results suggest that the core microbiota in the cecal microbiome was predominantly maintained after switching to the HG diet.

We also closely examined the 435 metagenome-assembled genomes (MAGs) (completeness of ≥80% and contamination of ≤10%) reconstructed from the metagenomic data of the four groups. The taxonomic affiliation of these MAGs was identified via GTDB-Tk, and the prevalent MAGs classified at the class level were *Clostridia* (59.3%), *Bacteroidia* (21.8%), and *Bacilli* (7.1%), while the top genera were *Alistipes* (5.5%), CAG:110 (*Firmicutes* bacterium) (4.6%), and *Ruminococcus* (3.7%) (Fig. 3C; Fig. S3 and Data Set S1). We further clustered these 435 MAGs into different co-occurrence modules using weighted correlation network analysis (WGCNA) and detected 20 dominant modules, with 9 to 43 MAGs in each module (Data Set S1). The genomic network exhibited a high degree of modularity, but 93.1% of edges accounted for only 5 of the 20 total modules (Fig. 3D). Moreover, these 5 modules, which contained MAGs mostly from *Oscillospirales* (41.4%) and *Bacteroidales* (30%), were all positively correlated with the hay diet and negatively correlated with HG diet feeding (defined as "hay dependent") (Fig. 3D). The time-dependent characteristics previously detected in the cecal microbiome were confirmed by revealing the relationship of the modules with fermentation parameters and separate feeding times. For example, module 16 was positively associated with HG diet feeding on day 7; modules 15 and 20 were positively associated with day 14; and modules 2, 12, and 14, which formed a large cluster of MAGs, representing MAGs from the *Alistipes*-affiliated populations MAG441 (*A. senegalensis*) and MAG587 (*Alistipes putredinis*), the *Bifidobacterium*-affiliated population MAG284 (*B. pseudolongum*), the butyrate-producing bacteria MAG536 (*Butyricimonas virosa*) and MAG456 (*Anaerostipes hadrus*), MAG292 (*Akkermansia muciniphila*), MAG396 (*Ruminococcus bromii*), and MAG51 (*Sharpea azabuensis*), were positively associated with day 28 (Fig. 3D). These results further support the adaptational response of the hindgut microbiome to the HG diet at the genomic level.

**Adaptation strategies of targeted functional groups in the cecal microbiome.** To determine the effect of the HG diet on microbial functions, the functional gene abundance involved in carbon and nitrogen metabolisms was closely examined using metagenomic data generated from the cecum samples of each sheep fed the hay and HG diets (Fig. 4; Fig. S4A and Data Set S2). By mapping the KEGG orthologous (KO) genes involved in carbohydrate metabolism, we found that the abundance of phosphate acetyltransferase (*pta*; K00625), the enzyme that catalyzes acetyl-CoA conversion to acetate, was increased in the HG groups. However, no significant differences were found in the process of acetyl-CoA conversion to butyrate (Fig. 4A). The metabolism of $CO_2$ was mainly associated with the Wood-Ljungdahl pathway to produce acetyl-CoA, and the abundance of enzymes involved in the methanogenesis pathway was reduced, including the methyl-coenzyme M reductase (*mcrABCD*), a central enzyme that catalyzes the final

**FIG 3** Legend (Continued)
the relative abundances of 301 core species among the four groups standardized using the *Z*-score method. Different-colored squares representing the core species were classified into different phyla and genera. (C) Taxonomic composition of the 435 MAGs at the class level, with ranks ordered from top to bottom by their decreasing proportions among the MAG collection. The six most frequently observed taxa are shown in the key. (D) Co-occurrence network in the cecal microbiome where MAGs (nodes) are colored according to WGCNA modules (named 1 to 20). Edges represent the positive or negative correlations selected based on the thresholds of a Spearman rank correlation coefficient of >0.85 and a *P* value of <0.01. The shading in the network represents the collection of most MAGs in the module. The heat map on the right shows the correlation between MAG modules and traits involved in diet, feeding time points, and fermentation parameters (Pearson's significance *P* value of <0.05). The taxonomic classification of the five modules ("hay-dependent") and the standardized TPM values of the six modules ("time-dependent") (MAGs with adjusted *P* values of <0.05 among the four groups were selected) are shown at the bottom.

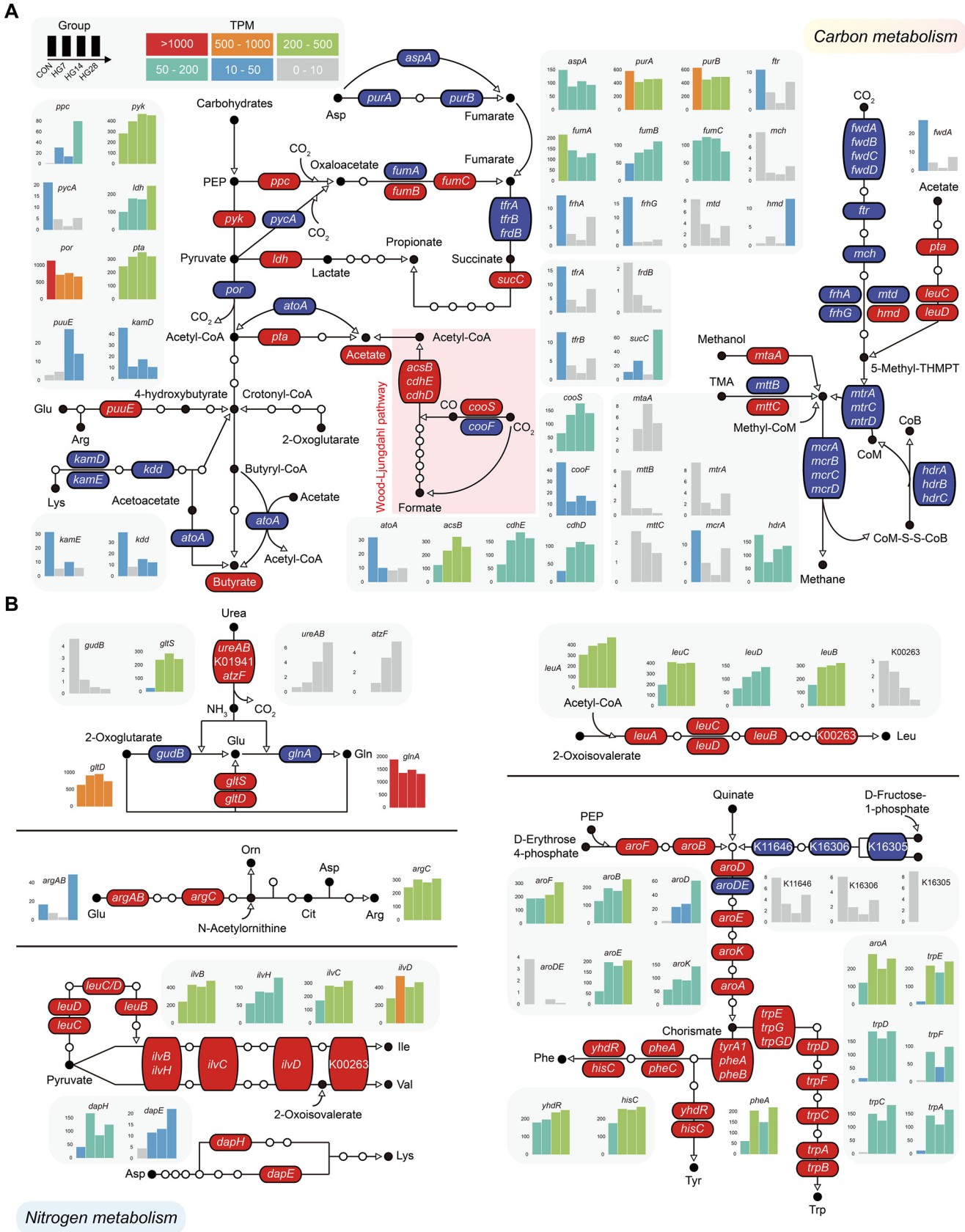

**FIG 4** Carbon and nitrogen metabolism among the four groups. (A) Metabolic pathways for VFA and methane production by microbial conversion from carbohydrates. (B) Metabolic pathways of urea metabolism and amino acid synthesis representing the utilization of ammonia by bacteria. Significantly

step of methanogenesis (Fig. 4A). We then focused on subsets of KO genes that corresponded to pyruvate biosynthesis and metabolism (Fig. S4B), which were supported by pyruvate kinase (*pyk*; K00873) and pyruvate-ferredoxin/flavodoxin oxidoreductase (*por*; K03737), the two key enzymes of VFAs produced from pyruvate conversion to acetyl-CoA. They were significantly up- and downregulated in the HG groups compared with the CON group, respectively. By identifying the taxonomic distribution of the two KOs based on the abundance of microbial genes, we found that *Ruminococcus bromii* was dominant on day 7 of the diet shift, while members of *Firmicutes* bacterium CAG:110 and *Prevotella ruminicola* were both dominant on day 14, and *Akkermansia muciniphila* was dominant on day 28 (Fig. S4B and C). These results indicate that the carbon metabolism of the cecum microbiome was shifted by the introduction of the HG diet and adapted over the feeding time.

By examining the relative abundance of genes in the pathway, we found that enzymes involved in amino acid biosynthesis, such as glutamate, lysine, leucine, and phenylalanine, had higher gene abundances in HG-fed animals (Fig. 4B). These were coincident with the exhibited varying abundances of urease enzymes hydrolyzing urea to ammonia and $CO_2$ (K14048 [UreAB], K01941 [urea carboxylase], and K01457 [allophanate hydrolase]). The ammonia generated by these steps served as the precursor for the synthesis of microbial proteins, mainly via the glutamate pathway for microbial growth. Members of the *Lachnoclostridium* spp., *Clostridium* spp., and *Roseburia* spp. showed an increased abundance of urease-containing genes in the HG groups compared with that in the CON group (Fig. S4C). These results indicated that the improved metabolic ability to use urea was a prevailing feature of the HG-fed animals across the feeding time, potentially owing to the HG diet driving the enhanced carbohydrate-hydrolytic enzyme activity of microbes under shifted environmental conditions in the hindgut.

**Relationships between epithelial gene expression and mucosa-associated microbiota.** Because the concentration of VFAs, the metabolic pathway of pyruvate, and urea metabolism significantly changed in the HG groups compared with the CON group, we then investigated the potential effect of altered microbiome homeostasis on the gene expression of the mucosal tissue (Table S2E). Hence, the relative mRNA expression levels of genes related to VFA absorption and metabolism, cytokines, and tight junction proteins were determined among the four groups (Fig. 5A; Tables S3A and B). We did not observe significant differences in the mRNA abundances of genes related to VFA absorption or genes for intracellular pH regulation from epithelial cells, such as *NHE1*, *NHE2*, *NHE3*, *DRA*, *MCT1*, and *MCT4*. Similarly, VFA metabolism in the epithelium also showed that genes associated with ketogenesis were not affected by grain inclusion (e.g., *BDH1*, *BDH2*, and *HMGCL*), but the mRNA abundance of *HMGCS1* appeared to be lower for HG-fed animals than for hay-fed animals; it was downregulated on day 28, which functioned as a catalyzer for butyrate metabolism for cholesterol synthesis. The cholesterol concentration in cells is linked to cellular inflammation and membrane permeability. Interleukin-6 (IL-6) is a proinflammatory cytokine that was upregulated in the cecum epithelial cells after the first week of the HG diet, and the anti-inflammatory cytokine IL-10 was downregulated on days 7 and 28. The downregulation of Toll-like receptor 3 (TLR-3) is related to pathogen-derived inflammatory responses on days 7 and 28, which may be impacted by changes in the mucosa-associated microbiota during the diet shift. In addition, high expression levels of both pro- and anti-inflammatory cytokines were detected on day 14, suggesting that the host might respond to abnormal homeostasis affected by the HG diet. Furthermore, the tight junction protein Claudin-1 was downregulated on day 28, and Claudin-4 was up-

**FIG 4** Legend (Continued)
different genes encoding enzymes involved in the pathways and VFA concentrations (acetate and butyrate) are shown in red (increase in HG-fed animals) and blue (decrease in HG-fed animals). Modules with no differences in the pathway are hidden by white circles. Distribution bar plots are colored according to the gene abundance as transcripts per million reads mapped (TPM). Detailed information on these 93 enzymes is presented in Data Set S2 in the supplemental material. PEP, phosphoenolpyruvate; TMA, trimethylamine; CoM-SS-CoB, heterodisulfide coenzyme M coenzyme B; THMPT, tetrahydromethanopterin.

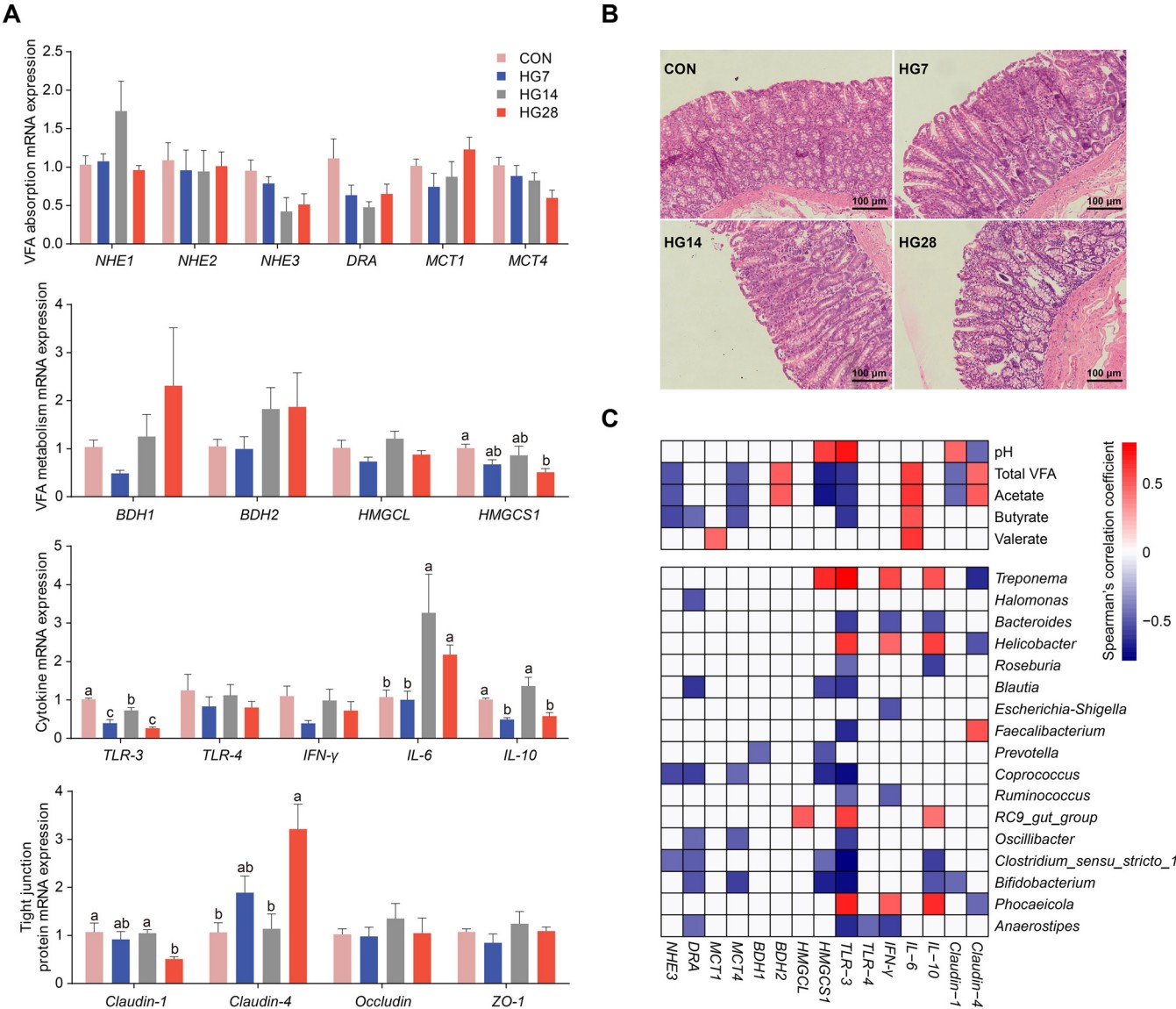

**FIG 5** Morphology and relative mRNA expression of the cecal epithelium. (A) mRNA abundances of genes involved in VFA absorption and metabolism, cytokines, and tight junction proteins among the four groups. Different letters represent significant differences among the four groups (P value of <0.05 by a Wilcoxon rank sum test). (B) Light microscopy cross section of cecum epithelial tissue in sheep fed the two diets. (C) mRNA expression of genes linked to cecum fermentation parameters and mucosa-associated microbiota. Only those that were significantly correlated are shown. The color of the cells represents Spearman's correlation coefficient (red, positive correlation; blue, negative correlation).

regulated on days 7 and 28, but differences in other proteins, such as occludin and ZO-1, were not detected. These results indicate that the HG diet shift may affect the function of the cecal epithelium and induce the epithelial inflammatory response, as evidenced by light microscopy cross-section results of the sloughing cecal epithelium surface from the HG-fed animals (Fig. 5B).

We then examined the association between cecal fermentation parameters and epithelial gene expression and found that the VFA absorption-related genes *NHE3* and *MCT4* negatively correlated with the concentrations of total VFAs, acetate, and butyrate (Fig. 5C). The VFA metabolism-related gene *BDH2* positively correlated with the concentration of total VFAs and acetate. However, the mRNA abundance of *HMGCS1* negatively correlated with the concentration of total VFAs and acetate and positively correlated with pH. *TLR-3* positively correlated with pH and negatively correlated with the concentrations of total VFAs, acetate, and butyrate, while *IL-6* positively correlated with the VFA concentration, including acetate, butyrate, and valerate. A reverse trend was observed

for *Claudin-1* and *Claudin-4* in correlation with the pH and the concentrations of total VFAs and acetate.

Epithelial gene expression was then linked to the mucosa-associated microbiota, and the results showed that the VFA absorption-related gene *DRA* was negatively correlated with both butyrate and acetate producers, including members of genera *Anaerostipes*, *Coprococcus*, *Bifidobacterium*, and *Blautia* (Fig. 5C). *TLR-3* was significantly correlated with most microbial genera. *TLR-3* controls the host's inflammatory response against pathogens, and we observed a positive correlation between *TLR-3* and pathogen-rich genera such as *Treponema*, *Helicobacter*, and *Phocaeicola*. Other anti-inflammatory cytokines, such as interferon gamma (IFN-$\gamma$) and IL-10, were positively correlated with these three genera, suggesting that the microbial community is critical to host adaptive immunity. These results indicate that perturbation of the hindgut microbiota alters the host immune system to modulate inflammation.

## DISCUSSION

In this study, we sampled sheep cecum at distinct time points, applying 16S rRNA gene and shotgun metagenomic sequencing to provide insight into the hindgut microbial adaptation response to the challenge of a diet shift. After switching from the hay to the HG diet, we found that the fermentation pattern of the hindgut changed, with a decrease in pH and an increase in the total VFA concentration. Accumulated acetate and butyrate were the key factors for the increase in the concentration of total VFAs. In addition, after feeding the HG diet for 4 weeks, acetate and butyrate concentrations continually increased, and the proportion of butyrate was steady at 10.8%. Similar results were also reported in a previous study in which we observed that rumen pH was decreased and the concentration of butyrate was increased in the HG group compared to the CON group (11). However, the concentration of acetate was not significantly changed in the rumen, which is likely related to the stronger buffering capacity of the rumen environment due to factors such as saliva and protozoa, which are lacking in the hindgut (9).

We then further examined the microbial composition, its role in VFA production, and epithelial function, which may have been affected by the changed concentration of VFAs. Not surprisingly, we found that the control and the three treatment groups separated significantly in both the digesta and mucosa, and a clear distinction was also observed among the three HG groups. Interestingly, the HG diet particularly contributed to a greater shift in digesta samples than in mucosa samples, which is likely related to the microbes adhering to the intestinal mucosa being genetically controlled by the host through direct regulation via host-microbe interactions (20). We also identified that the richness of the hindgut microbiota was reduced after the diet shift, and the distance was the lowest on day 14 compared to the CON group, suggesting progressive microbial adaptation under the environment perturbation.

After analyzing the structure of the hindgut microbiome, we found that acetate and butyrate were negatively correlated with microbial community diversity, and acetate and butyrate producers were prominent within the HG-fed animals. These results indicate that the HG diet may transform the hindgut microbiota into a fermentation pattern for acetate and butyrate production. The core microbiota in the cecal microbiome was predominantly maintained after HG diet feeding, which may indicate that the changes in the cecal microbiome mainly occurred in microbial abundance. By examining the specific populations of the cecal microbial community at the species and genomic levels, we found that these taxa exhibited a significant time-dependent characteristic. In the first 2 weeks, the excess fermentable carbohydrates in the foregut flowed downward, inducing the proliferation of bacteria related to polysaccharide degradation in the cecum, such as *Prevotella* (*P. ruminicola* and *Prevotella* sp. KHP1)- and *Ruminococcus* (*Ruminococcus bromii* and *Ruthenibacterium lactatiformans*)-affiliated populations. Regarding the fourth week, the homeostasis of the cecal microbiome may be out of balance because some microorganisms relevant to dysbiosis and disease were enriched, such as *Alistipes*-affiliated

populations belonging to *A. senegalensis* (MAG441) and *A. putredinis* (MAG587). However, we also observed that many probiotics, including *Bifidobacterium*-affiliated populations (*B. pseudolongum* and *B. merycicum*) and *Akkermansia muciniphila* (MAG292), were significantly enriched on day 28. The prevalent aggregation of these bacteria may represent the repair of damage caused by the challenge of long-term HG diet feeding. While a decrease was revealed in hindgut microbial richness caused by the HG diet, we found an increase in taxa of acetate and butyrate producers, which may represent an alteration of trophic levels of carbon processing under this changed environment in the hindgut.

Thus, we then focused on the microbial genes in the carbohydrate metabolism pathway and found that the metabolism of $CO_2$ was mainly associated with the Wood-Ljungdahl pathway to produce acetyl-CoA. Interestingly, there were no significant differences in the abundances of enzymes involved in the acetyl-CoA pathway in butyrate production. Since the acetyl-CoA pathway is present in the majority of butyrate producers (21), this may help explain why the concentration and proportion of butyrate in the HG groups increased. We also found that *pyk* and *por* were significantly up- and downregulated in the pathway of pyruvate biosynthesis and metabolism in the cecum of sheep from the HG groups, respectively. These results agree with the idea that the pyruvate metabolism of the hindgut microbiome shifted when the HG diet was introduced. Intriguingly, the composition of microbes encoding *pyk* and *por* was altered by the feeding time. *Ruminococcus bromii* is a keystone species for starch degradation (22), and it was the most prevalent species on day 7; both *Firmicutes* bacterium CAG:110 and *Prevotella ruminicola* were the most prevalent species on day 14; and *Akkermansia muciniphila* was dominant on day 28. *Firmicutes* bacterium CAG:110 is a newly discovered species, and a recent study found that it was potentially associated with high feed efficiency in swine cecum microbiota (23). *Prevotella ruminicola* is an anaerobic bacterium that contributes to hemicellulose utilization and can degrade proteins and noncellulosic polysaccharides such as starch (24). *Akkermansia muciniphila* is an acetate producer derived from pyruvate via acetyl-CoA, which is believed to have anti-inflammatory effects on humans (25). It is possible to conclude that the carbon metabolism of the hindgut microbiome was regulated by different microbes during HG diet feeding. It is also notable that there was an improved metabolic ability to utilize urea, and across the experimental feeding times, nitrogen metabolism-related enzymes were universally enhanced in the HG-fed animals. This diet shift clearly alters the available substrates that flow into the hindgut, changing from a predominantly fiber substrate to a fermentable carbohydrate substrate such as starch.

The absorption of VFAs is an important consideration for the regulation of lumen pH through gut epithelial cells (26). We did not find differences in the mRNA abundances of VFA absorption-related genes among the four groups, and the VFA metabolism-related gene *HMGCS1* was downregulated on day 28, which was positively correlated with the cecum pH. This may lead to the accumulation of cecum VFAs and a shift of the lumen pH. Dietary composition has been shown to affect colonization by pathogenic species (27). We revealed that the HG diet may cause an epithelial inflammatory response and the enrichment of pathogen-rich genera such as *Treponema*, *Helicobacter*, and *Phocaeicola* in the mucosa-associated microbiota. These genera are positively correlated with host gene expression associated with anti-inflammatory cytokines, including *TLR-3*, *IFN-γ*, and *IL-10*, in epithelial cells. These results indicate that a diet shift profoundly affects the composition of the microbiota and the concentration of VFAs in the hindgut, with repercussions for immunity and susceptibility to pathogenic infection of the host.

In conclusion, this study highlighted that the introduction of the HG diet leads to the perturbation of the hindgut microbial composition and metabolism and results in mucosal injuries in sheep. Using an integrated analysis, we illustrated that the increased abundance of acetate and butyrate producers in the cecum after HG diet feeding induces the accumulation of acetate and butyrate, which are important energy sources for animals. However, the composition and metabolism of the cecal microbiome changed during SARA, and mucosa-associated microorganisms provoked an

epithelial immune response and resulted in mucosal injuries. Interestingly, some probiotics were enriched on day 28, which may imply that when hindgut homeostasis is unbalanced, the microbial community will try to repair itself, or it may be some kind of manipulation by the host. If the diet structure can be adjusted within 4 weeks, it could probably improve hindgut SARA in time and restore animal health. In addition, we also indicate that the essential metabolic functions in the cecal microbial community are controlled by a few core species, which are predominantly maintained after HG diet feeding, but their abundance fluctuates. Therefore, investigating their living preferences through genomic research and then modifying the nutritional composition to target their abundance may ultimately regulate the metabolism of the hindgut microbiota in keeping with our needs. Overall, these findings offer a comprehensive profile of the cecal microbial structure and the time course of adaptation during HG diet feeding. They also help to further elucidate the etiology of hindgut disorders in ruminants. Future work should focus on exploring the microbiota-host variant dynamics in the hindgut of ruminants under a highly fermentable diet feeding regimen.

## MATERIALS AND METHODS

**Animals and experimental design.** A total of 20 male Hu sheep (*Ovis aries*) (180 days; 25.60 $\pm$ 0.41 kg) were used in this study. The experiment was conducted at the animal station of Nanjing Agricultural University. Animals were randomly assigned to four groups ($n = 5$) and kept in individual cages (1.2 by 1.4 m). At first, the animals were fed the same hay diet for 4 weeks. After the adaptation period, the animals in the three HG groups were fed the HG diet for 4 days (the grain concentration was increased by 15% per day) until the proportion of grain reached 60%, while the CON group continued to be fed the hay diet (see Table S4 in the supplemental material). In the experimental period, the CON group was fed the hay diet for 28 days, and the HG groups received the HG diet for 7 days (HG7), 14 days (HG14), and 28 days (HG28). All animals were fed based on 3.5% of their body weight at 08:30 and 16:30. The body weights of these sheep were measured on the first day of every week before the morning feeding, and the residuals removed were collected daily.

**Sample collection.** On the last day of the experimental period, the animals were slaughtered 4 to 5 h after the morning feeding. The digestive tract was dissected, and the cecum was immediately separated. Cecal digesta samples were collected, and the pH was immediately measured using a pH meter (catalog no. HI 9024C; Hanna Instruments, Woonsocket, RI, USA). Next, each sample was diluted with distilled water, and the mixture was then centrifuged at 2,000 $\times$ $g$ for 10 min to separate the solid residues and liquid. The supernatant fluid was mixed with 25% (wt/vol) metaphosphoric acid (5 ml supernatant fluid and 1 ml metaphosphoric acid) and stored at $-20°C$ for quantifying VFA concentrations by gas chromatography (GC-14B; Shimadzu, Japan). The cecal mucosal samples were scraped using a sterile slide and then stored at $-80°C$ for subsequent microbial DNA extraction. Cecal epithelium tissues were collected, washed with ice-cold phosphate-buffered saline, and then divided into two parts. One part of the samples was cut into 0.4- by 0.4-cm sections and then kept at $-80°C$ for subsequent RNA extraction. The other part of the samples was fixed in 4% paraformaldehyde for histomorphometric microscopy analysis. All animal-specific procedures were approved and authorized by the Nanjing Agricultural University Institutional Animal Care and Use Committee (no. SYXK-2017-0027).

**DNA extraction.** All digesta and mucosa samples were thawed at 4°C, and ~200 mg per sample was placed into 2-ml centrifuge tubes with 0.2 g of bead powder. The treatment solution was added according to the instructions of the DNA isolation kit (E.Z.N.A. soil DNA kit; Omega Bio-Tek, USA). The samples were then placed in a bead-beating grinder to lyse the samples. DNA was extracted from each sample according to the manufacturer's instructions. The concentration of DNA was determined by a Nanodrop ND-1000 instrument (Thermo Scientific, USA), and agarose gel electrophoresis was used to evaluate the quality of the DNA.

**16S rRNA gene and shotgun metagenomic sequencing.** We used universal primers (341F [5'-CCTAYGGGRBGCASCAG-3'] and 806R [5'-GGACTACNNGGGTATCTAAT-3']) to amplify the V3-V4 regions of the 16S rRNA gene with a 6-bp barcode unique to each sample. The resulting amplicons were purified using a QIAquick PCR purification kit (Qiagen, Hilden, Germany) and subjected to library construction according to the Illumina manufacturer's instructions. All libraries were sequenced using an Illumina MiSeq PE-250 platform. The DNA samples from cecal digesta were additionally subjected to shotgun metagenomic sequencing. The metagenomic DNA library was constructed according to the Illumina manufacturer's instructions and then sequenced on an Illumina NovaSeq platform.

**16S rRNA analysis.** Trimmomatic (v.0.33) (28) was used to remove adapters and low-quality sequences. The data were processed using a QIIME pipeline (v.1.9.0) (29). Sequences were clustered into operational taxonomic units (OTUs) based on a sequence similarity level of 97% using UPARSE (v.7.1) (30). Taxonomic assignment of the OTUs was performed by mapping the representative sequences to the SILVA 16S rRNA database (v.11.9) using the Bayesian classifier of the Ribosomal Database Project (31). The archaeal sequences were deleted, and phylogenetic analysis was subsequently standardized based on the sequencing depth. The number of sequences in the sample with the lowest sequencing depth was used as the standardized sampling size to recluster the OTUs of all samples. The numbers of standardized sequences in the cecal digesta and mucosa samples were 29,154 and 29,391, respectively.

Microbial profiles were evaluated by the sequencing and clustering of 16S rRNA gene sequences into OTUs; in cecal digesta and mucosa communities, 1,849 and 1,723 OTUs were identified, respectively.

**Metagenomic analysis.** Illumina sequencing generated a total of 421.3 Gb of raw data, with an average of 140.4 million reads per sample. All adapters and low-quality sequences were trimmed using Trimmomatic (v.0.33) (28). Afterward, BWA-MEM (v.0.7.17) (32) was used to remove the reads that mapped to human, sheep, and plant (including corn, wheat, rice, soybean, and alfalfa) genomes, and 317.6 Gb (average of 15.9 Gb per sample) of high-quality reads were obtained. These cleaned sequences were used as the input data for the metagenomic assembly. We assembled the sequences from each sample using MEGAHIT (v.1.1.1) (33). An average of 74.3% of the cleaned reads were used per sample and finally assembled into a total of 6.7 million contigs with a length exceeding 500 bp. The average $N_{50}$ length was 6,245. Contigs in each sample were selected to predict open reading frames (ORFs) using Prodigal (v.2.6.3) (34) with the parameter option "-p meta." ORFs with a length of <100 bp were deleted. All the predicted ORFs were combined to generate the nonredundant microbial gene catalog using CD-HIT (v.4.6.7) (35), with criteria of ≥95% identity and ≥90% overlap. We finally obtained 8,010,225 nonredundant genes with an average length of 661.6 bp. Functional assignments of the microbial gene catalog were performed using DIAMOND (v.0.9.22) (36) based on the BLASTP procedure against the NCBI-NR and KEGG (v.90.0) (37) protein sequence databases by obtaining the best match under the criterion of an E value of <1e−5. Abundance profiles were evaluated by mapping the cleaned reads per sample back to the gene catalog using BWA-MEM (v.0.7.17) (32) and calculated in transcripts per million (TPM) (alignment length of ≥50 bp and sequence identity of >95%).

**Metagenomic binning.** Metagenomic contigs were binned into genomes using MaxBin (v.2.2.4) (38), MetaBAT2 (v.2.11.1) (39), and CONCOCT (v.0.4.0) (40) binning software. The DAS tool (v.1.1.1) (41) was then used to integrate all the MAGs. We finally assembled a total of 3,824 MAGs. Next, all the MAGs were dereplicated with a 99% average nucleotide identity (ANI) cutoff using dRep (v.2.5.4) (42), and 1,194 nonredundant MAGs were obtained. The completeness and contamination of all MAGs were assessed using CheckM (v.1.0.7) (43) based on the lineage_wf workflow. Only the MAGs that met or exceeded the thresholds of ≥80% completeness and ≤10% contamination were employed for downstream analysis. All MAGs were taxonomically annotated using GTDB-Tk (v.0.1.6) (44) based on the Genome Taxonomy Database (http://gtdb.ecogenomic.org/). The abundance of MAGs in each sample was estimated using metaWRAP (v.1.3) (45) with a "quant_bins" module. A phylogenetic tree of the 435 MAGs was built using PhyloPhlAn (v.1.0) (46) by aligning the individual proteins from the protein sets recovered from these genomes and was visually inspected using iTol (v.4.3.1) (47).

**RNA extraction and real-time PCR.** The extraction of RNA from the epithelial tissue was performed using a TRIzol kit (TaKaRa Bio, Otsu, Japan) according to the manufacturer's instructions. The RNA concentration was quantified, and RNA quality was evaluated using a Nanodrop ND-1000 instrument (Thermo Scientific, USA). The concentration of each RNA sample was subsequently adjusted to 500 ng/ml, and the total RNA was reverse transcribed using a PrimeScript RT reagent kit with gDNA Eraser (TaKaRa Bio, Otsu, Japan). Table S3B shows all the primers used in this study. All primers were synthesized by Invitrogen Life Technologies (Shanghai, China). Target gene expression was quantified using the ABI 7300 real-time PCR system (Applied Biosystems, Foster City, CA, USA), according to a previously described program (15).

**Statistical analyses.** The Kruskal-Wallis rank-sum test was used to detect the significance of the experimental data for the four groups, and the Wilcoxon rank sum test was then used to compare the two groups. All $P$ values were calculated using kruskal.test and wilcox.test in R with the parameter "paired = FALSE," and p.adjust with the parameter "method = fdr" was then used to correct the false discovery rate (FDR). An adjusted $P$ value of <0.05 (FDR < 0.05) was considered statistically significant. We calculated the correlations using the Hmisc package of R based on the Spearman correlation test with an asymptotic measure-specific $P$ value. Co-occurrence modules were inferred using weighted correlation network analysis with the R package WGCNA (v.1.69) (48), and the network was graphed using Gephi (v.0.9.2) (49). The complete list of MAGs and their module organization are available in Data Set S1.

**Data availability.** Raw sequence reads for all samples are accessible under the European Nucleotide Archive (ENA) project accession no. PRJNA702231. All 435 MAGs utilized in this study have been deposited in Figshare (https://doi.org/10.6084/m9.figshare.14453094).

## SUPPLEMENTAL MATERIAL

Supplemental material is available online only.

**FIG S1**, TIF file, 2.3 MB.
**FIG S2**, TIF file, 2.2 MB.
**FIG S3**, TIF file, 2 MB.
**FIG S4**, TIF file, 2.8 MB.
**TABLE S1**, DOCX file, 0.01 MB.
**TABLE S2**, XLSX file, 0.03 MB.
**TABLE S3**, XLSX file, 0.01 MB.
**TABLE S4**, DOCX file, 0.01 MB.
**DATA SET S1**, XLSX file, 0.1 MB.
**DATA SET S2**, XLSX file, 0.02 MB.

## ACKNOWLEDGMENTS

This work was supported by the Jiangsu Agriculture Science and Technology Innovation Fund [CX(19)1006] and the Six-Talent Peaks Project in Jiangsu Province (grant no. NY-050), China.

We are grateful for the support of the high-performance computing platform of the Bioinformatics Center, Nanjing Agricultural University.

S.M. designed and supervised the study. F.X., L.X., and Y.W. performed the experiments. F.X. and Y.W. analyzed and interpreted the data. F.X. wrote the manuscript. All authors read, revised, and approved the final manuscript.

We declare no financial conflicts of interest.

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
