## [Reviewer comments · mSystems]

Metagenomic sequencing reveals high-grain feeding alters the composition and metabolism of cecal microbiota and induces cecal mucosal injury in sheep

Fei Xie, Lei Xu, Yue Wang, and Shengyong Mao

Corresponding Author(s): Shengyong Mao, Nanjing Agricultural University

Review Timeline:

Submission Date:	July 13, 2021
Editorial Decision:	August 6, 2021
Revision Received:	August 28, 2021
Accepted:	September 12, 2021

Editor: Suzanne Ishaq

Reviewer(s): Disclosure of reviewer identity is with reference to reviewer comments included in decision letter(s). The following individuals involved in review of your submission have agreed to reveal their identity: Lianmin Chen (Reviewer #1); Kimberly Anne Dill-McFarland (Reviewer #2)

Transaction Report:

DOI: <https://doi.org/10.1128/mSystems.00915-21>

August 6, 2021

Prof. Shengyong Mao
Nanjing Agricultural University
Nanjing
China

Re: mSystems00915-21 (Metagenomic sequencing reveals high-grain feeding alters the composition and metabolism of cecal microbiota and induces cecal mucosal injury in sheep)

Dear Prof. Shengyong Mao:

Thank you for submitting your manuscript to mSystems. We have completed our review and I am pleased to inform you that, in principle, we expect to accept it for publication in mSystems. However, acceptance will not be final until you have adequately addressed the reviewer comments.

Preparing Revision Guidelines

For complete guidelines on revision requirements for your article type, please see the journal Article Types requirement at <https://journals.asm.org/journal/mSystems/article-types>. **Submissions of a paper that does not conform to mSystems guidelines will delay acceptance of your manuscript.**

Sincerely,

Suzanne Ishaq

Editor, mSystems

Journals Department
Reviewer comments:

Reviewer #1 (Comments for the Author):

In this manuscript, Xie and colleagues present a valuable data source of cecal microbiome profiles in SARA condition. They have compared cecal microbiome differences between normal and SARA conditions and further checked potential interactions between the microbiome and mucosal epithelial gene expression. Major and minor comments are listed below:

Major:

1. The authors have applied both 16S-RNA and metagenomic sequencing. In this case, I would suggest only go for metagenomic profiles (MGS) as its resolution is much higher and you can easily check species level instead of genus level.
2. Regarding the core microbiome part, I suggest changing it to species level by using species abundance profiles generated from MGS. Do such analysis based on OUTs generated from 16S-RNA is not that attractive when metagenomic data is available, as the resolution of MGS is much higher.
3. L511, It's reasonable to use rank-based statistics, but it's not correct to use "paired comparison" as your samples (animals) were not paired (L418 the animals were slaughtered... so not the same animals have been followed for many times), this has to be changed to an un-paired approach e.g., un-paired Wilcoxon test.
4. As the sample size in this study was limited (i.e., 5 samples per group), the authors didn't apply strict FDR correction. This has to be acknowledged that there might be a lot of FP in their results.

Minor:

1. Fig2B&E, box plots with dots need to be fixed as all individual dots from different groups were crushed together.

2. grammar mistakes need to be checked throughout the text e.g., L293 should be "phylogenetics" instead of "phylogenetic".

Reviewer #2 (Comments for the Author):

In this study, the authors investigate the impacts of a high-grain diet on sheep cecal microbiota and nutritional environment. With a combination of microbial (16S OTUs, shotgun metagenomics), metabolite (VFA), and host (rtPCR, microscopy) techniques, the authors provide a comprehensive exploration. Overall, the manuscript is well-written, and the conclusions drawn are supported by the data. The major comments I have are regarding figures. They are large and should be considerably improved for readability and comprehension. Please see specific comments below.

Line 39: Unclear what "the enzymes" refers to

Line 395: Please expand on the importance of the results found. The authors report improved weight gain with a HG diet but potentially negative outcomes from mucosal injury, pH, and microbiota changes. How might the results inform feeding practices to balance these outcomes and potentially prevent acidosis? Are there target species or pathways for pre/probiotic development?

Line 416: Should "residue" be "residual" as in residual feed intake?

Line 461: What standardization was performed? Rarefaction?

Line 153: It is important to note that the directionality of this change is unknown. Both diet → microbe → pH and diet → pH → microbe are potential modes of action

Line 302: Why was WGCNA done on only core taxa? With the numerous differentially abundant OTUs, using the full data set would yield modules more strongly associated with diet and other parameters of interest. Further WGCNA reduces complexity, thus allowing less harsh FDR correction for downstream statistical test. Since authors found and discuss in detail many gene-level results, the modules do not appear to add to the results. Honestly, I suggest removing WGCNA entirely.

Figures overall: Coloring should be consistent for CON and HG groups throughout, particularly as red is the control group in Figure 1 but is HG28 in Figure 2.

Figure 3: While this figure is impressive looking, it adds very little to the narrative relative to its size. Suggest moving to supplemental and perhaps just highlighting the taxonomic bar plot and/or heatmap of the labeled MAGs in the main text.

Figure S4: Comprehension would be improved by clustering rows within the pathway groups by genes that increase/decreased with HG

Figure 4: TPM barplots cluster the message and are redundant with red/blue coloring in the pathways. Suggest moving to supplemental, in alphabetical order to easily find specific genes of interest in the pathways

Figure 4: Many of the VFAs in the described pathway were measured in this system. Thus, the red/blue coloring could be expanded to include significant changes in VFAs, thus highlighting which changes in gene abundance may result in measurable changes in the gut.

Figure 6AB: The purpose of WGCNA is to collapse co-expressed genes into modules. Thus, showing correlations between genes not in the same module does not convey useful information. Suggest removing these unreadable networks.

Figure 6: Using color words to denote modules is confusing when those colors are not used (D) and when those colors are used to denote taxonomy instead in another panel (E)

Figure 6D: It would be clearer to include the module-module correlations in the heatmap in (C)

Figure 6E: Why is one line jagged?

The point-by-point response to the reviewers' comments

Reviewers' comments

Reviewer #1:

In this manuscript, Xie and colleagues present a valuable data source of cecal microbiome profiles in SARA condition. They have compared cecal microbiome differences between normal and SARA conditions and further checked potential interactions between the microbiome and mucosal epithelial gene expression. Major and minor comments are listed below:

R: Thank you very much for reviewing our manuscript and giving these constructive suggestions. In the revision, we have addressed all these issues according to your suggestions.

Major:

1. The authors have applied both 16S-RNA and metagenomic sequencing. In this case, I would suggest only go for metagenomic profiles (MGS) as its resolution is much higher and you can easily check species level instead of genus level.

R: Thank you for this important suggestion. We agreed that metagenomic profiles are more accurate for classification. Thus, we have reanalyzed the results of the taxonomic composition and further focused on species and MAG levels (Lines 146-210). These results are shown in the redrawn Figure 3. Moreover, in this study, we used 16S rRNA sequencing for mucosal microorganisms due to current technical limitations. Thus, we have retained the comparison results of digesta and mucosa microbial diversity under the interference of the HG diet based on the 16S rRNA gene profile in lines 110-145.

2. Regarding the core microbiome part, I suggest changing it to species level by using species abundance profiles generated from MGS. Do such analysis based on OTUs generated from 16S-RNA is not that attractive when metagenomic data is available, as the resolution of MGS is much higher.

R: Thanks for your suggestion. Indeed, OTU-based core microbiome analysis is weak, because it is hard to trace the specific species. Thus, we have reanalyzed this part using the species abundance generated from MGS according to your suggestion. We also believed that the new results need to be combined as a part of the results of the taxonomic composition, so we have reorganized its position in the revision (lines 167-184, Figure 3B and Table S2D).

3. L511, It's reasonable to use rank-based statistics, but it's not correct to use "paired comparison" as your samples (animals) were not paired (L418 the animals were slaughtered... so not the same animals have been followed for many times), this has to be changed to an un-paired approach e.g., un-paired Wilcoxon test.

R: Thanks for pointing out this confusion. We have changed the statistical analysis of significance in this study by using Kruskal-Wallis rank-sum test (for four groups) and Wilcoxon rank-sum test (for two groups). We have also rephrased the descriptions in the *Materials and methods* to "The Kruskal-Wallis rank-sum test

was used to detect the significance of the experimental data of the four groups, and then the Wilcoxon rank-sum test was used to compare the two groups. All P-values were calculated using `kruskal.test` and `wilcox.test` in R with the parameter "paired = FALSE," and then `p.adjust` with the parameter "method = fdr" was used to correct the false discovery rate (FDR). The adjusted P-value < 0.05 (FDR < 0.05) was considered statistically significant." in lines 527-532.

4. As the sample size in this study was limited (i.e., 5 samples per group), the authors didn't apply strict FDR correction. This has to be acknowledged that there might be a lot of FP in their results.

R: Thanks for your comment. We have corrected the P-values in this study using the FDR method and described it in the *Materials and methods* "All P-values were calculated using `kruskal.test` and `wilcox.test` in R with the parameter "paired = FALSE," and then `p.adjust` with the parameter "method = fdr" was used to correct the false discovery rate (FDR). The adjusted P-value < 0.05 (FDR < 0.05) was considered statistically significant." in lines 529-532. According to the calculation formula of the FDR method, a large number of P-values are required as the basis for sorting. Therefore, the corrections in our study were only for the results of high-throughput sequencing.

Minor:

1. Fig2B&E, box plots with dots need to be fixed as all individual dots from different groups were crushed together.

R: Thanks for your suggestion. We have divided the dots into different groups in the revision (Figure 2B and 2E).

2. grammar mistakes need to be checked throughout the text e.g., L293 should be "phylogenetics" instead of "phylogenetic".

R: Thanks for this suggestion. We have rechecked the language and writing of this manuscript and marked the changes in the new version with a grey background.

Reviewer #2:

In this study, the authors investigate the impacts of a high-grain diet on sheep cecal microbiota and nutritional environment. With a combination of microbial (16S OTUs, shotgun metagenomics), metabolite (VFA), and host (rtPCR, microscopy) techniques, the authors provide a comprehensive exploration. Overall, the manuscript is well-written, and the conclusions drawn are supported by the data. The major comments I have are regarding figures. They are large and should be considerably improved for readability and comprehension. Please see specific comments below.

R: Thank you very much for reviewing our manuscript and giving these important comments. In the revision, we have addressed all these issues regarding figures and contents based on your suggestions.

Line 39: Unclear what "the enzymes" refers to

R: Sorry for this ambiguity. We have rephrased this sentence to *"Association analysis suggests that the perturbations of the cecal microbiome under the HG diet were closely linked to the variations in the two key enzymes that catalyze the conversion of pyruvate to acetyl-CoA and urease enzymes that hydrolyze urea into ammonia, alongside mucosal inflammatory responses."* in lines 25-29.

Line 395: Please expand on the importance of the results found. The authors report improved weight gain with a HG diet but potentially negative outcomes from mucosal injury, pH, and microbiota changes. How might the results inform feeding practices to balance these outcomes and potentially prevent acidosis? Are there target species or pathways for pre/probiotic development?

R: We thank you for this question. We fully agreed with reviewer's comments, which are very important for highlighting the significance and importance of this study. Therefore, we have added some summary and outlook in the conclusion *"Using integrated analysis, we illustrated that the increased abundance of acetate and butyrate producers in the cecum after HG diet feeding induces an accumulation of acetate and butyrate, which are important energy sources for animals. However, the composition and metabolism of the cecal microbiome changed during SARA, and mucosa-associated microorganisms provoked epithelial immune response and resulted in mucosal injuries. Interestingly, some probiotics were enriched on day 28, which may imply that when hindgut homeostasis is unbalanced, the microbial community will try to repair itself, or it may be some kind of manipulation by the host. If the diet structure can be adjusted within four weeks, it could probably improve hindgut SARA in time and restore animal health. In addition, we also indicate that the essential metabolic functions in the cecal microbial community are controlled by few core species, which were predominantly maintained after HG diet feeding, but their abundance has fluctuated. Therefore, investigating their living preferences through genomic research and then modifying the nutritional composition to target their abundance may ultimately regulate the metabolism of the hindgut microbiota in keeping with our needs."* in lines 393-408.

Line 416. Should "residue" be "residual" as in residual feed intake?

R: Thanks. We have replaced the word "residue" with "residual" in the revision (in line 426).

Line 461: What standardization was performed? Rarefaction?

R: Thanks for your comment. In this study, we used the lowest sequencing depth among all samples as the standardized sampling size to re-cluster OTU. We have added a description of this method "*Rarefaction curves approached asymptotes across all animals and treatments, implying that the sequencing depth allows for an effective standardization of biodiversity measures (Fig. S1B).*" in lines 115-117 and "*The archaeal sequences were deleted and phylogenetic analysis subsequently standardized based on the sequencing depth. The number of sequences in the sample with the lowest sequencing depth was used as the standardized sampling size to re-cluster the OTUs of all samples. The standardized sequence numbers in the cecal digesta and mucosa samples were 29,154 and 29,391, respectively. Microbial profiles were evaluated by the sequencing and clustering of 16S rRNA gene sequences into OTUs; in cecal digesta and mucosa communities, 1,849 and 1,723 OTUs were identified, respectively.*" in lines 470-477.

Line 153: It is important to note that the directionality of this change is unknown. Both diet -> microbe -> pH and diet -> pH -> microbe are potential modes of action

R: Thanks for pointing out this oversight. We have replaced the words "affected by" with "associated with" and revised this sentence to "..., and that this impact was likely associated with the variations in the pH and VFA concentration." in lines 128-129.

Line 302: Why was WGCNA done on only core taxa? With the numerous differentially abundant OTUs, using the full data set would yield modules more strongly associated with diet and other parameters of interest. Further WGCNA reduces complexity, thus allowing less harsh FDR correction for downstream statistical test. Since authors found and discuss in detail many gene-level results, the modules do not appear to add to the results. Honestly, I suggest removing WGCNA entirely.

R: We thank you for this constructive suggestion. We agreed that WGCNA can effectively cluster taxa into modules with the same situation to reduce complexity. Thus, we have reanalyzed the taxonomic data using WGCNA analysis based on 435 MAGs reconstructed from the metagenomic data of the four groups (because OTU is hard to trace specific species). We found that the network of these MAGs exhibits a high degree of modularity and time-dependent characteristic, which further supports the adaptational response of cecal microbiome to the HG diet. Thus, we have combined these results with the taxonomic composition in lines 191-208 (Figure 3D).

Figures overall: Coloring should be consistent for CON and HG groups throughout, particularly as red is the control group in Figure 1 but is HG28 in Figure 2.

R: Thanks for your important suggestions regarding the figures in our study. In the revision, we have unified the colors of the four groups in all figures.

Figure 3: While this figure is impressive looking, it adds very little to the narrative relative to its size. Suggest moving to supplemental and perhaps just highlighting the taxonomic bar plot and/or heatmap of the labeled MAGs in the main text.

R: Thanks for your suggestion. We agreed that the information of this figure is not enough to be a main figure. Thus, in the revision, we have merged the new results from the species and MAG analysis into Figure 3, and moved the genomic tree to the supplementary file based on your suggestion (Fig. S3).

Figure S4: Comprehension would be improved by clustering rows within the pathway groups by genes that increase/decreased with HG.

R: Thanks. This is a great idea and we have modified this figure as you suggested (Fig. S4A).

Figure 4: TPM barplots cluster the message and are redundant with red/blue coloring in the pathways. Suggest moving to supplemental, in alphabetical order to easy find specific genes of interest in the pathways.

R: We thank you for this suggestion. In this figure, we want to show that the changes of these enzymes in the pathways are related to the feeding time of the HG diet, because the effect is time-dependent, as we have shown in the microbial composition. Therefore, after careful discussion, we hope to add these bar-plots to better present their dynamic changes.

Figure 4: Many of the VFAs in the described pathway were measured in this system. Thus, the red/blue coloring could be expanded to include significant changes in VFAs, thus highlighting which changes in gene abundance may result in measurable changes in the gut.

R: Thanks. This is also a great idea and we have added the changes of acetate and butyrate concentrations to this figure (Figure 4A), which are the two significantly changing VFAs that we focused on in this study.

Figure 6AB: The purpose of WGCNA is to collapse co-expressed genes into modules. Thus, showing correlations between genes not in the same module does not convey useful information. Suggest removing these unreadable networks.

R: Thanks for your suggestion. We agreed that WGCNA is to cluster taxa into modules to reduce complexity. We have removed this result and the related figures based on your suggestion. Moreover, the new results based on 435 MAGs reconstructed from the metagenomic data were exhibited in Figure 3D.

Figure 6: Using color words to denote modules is confusing when those colors are not used (D) and when those colors are used to denote taxonomy instead in another panel (E)

Figure 6D: It would be clearer to include the module-module correlations in the heatmap in (C)

R: Thanks for pointing out this confusion. In the new analysis based on these MAGs, we have redefined the module name and used the heatmap to exhibit the correlations (Figure 3D).

Figure 6E: Why is one line jagged?

R: Sorry for this mistake. We have removed this part in the revision (Figure 3D).

September 12, 2021

Prof. Shengyong Mao
Nanjing Agricultural University
Nanjing
China

Re: mSystems00915-21R1 (Metagenomic sequencing reveals high-grain feeding alters the composition and metabolism of cecal microbiota and induces cecal mucosal injury in sheep)

Dear Prof. Shengyong Mao:

Your manuscript has been accepted, and I am forwarding it to the ASM Journals Department for publication. For your reference, ASM Journals' address is given below. Before it can be scheduled for publication, your manuscript will be checked by the mSystems senior production editor, Ellie Ghatineh, to make sure that all elements meet the technical requirements for publication. She will contact you if anything needs to be revised before copyediting and production can begin. Otherwise, you will be notified when your proofs are ready to be viewed.

As an open-access publication, mSystems receives no financial support from paid subscriptions and depends on authors' prompt payment of publication fees as soon as their articles are accepted. =

Publication Fees:

We recognize that the video files can become quite large, and so to avoid quality loss ASM suggests sending the video file via <https://www.wetransfer.com/>. When you have a final version of the video and the still ready to share, please send it to Ellie Ghatineh at eghatineh@asmusa.org.

Sincerely,

Suzanne Ishaq
Editor, mSystems

Journals Department
Table S3: Accept
Fig. S4: Accept
Fig. S1: Accept
Data Set S1: Accept
Table S4: Accept
Table S1: Accept
Fig. S3: Accept
Data Set S2: Accept
Table S2: Accept
Fig. S2: Accept